# Nutritional Modulation of the Gut–Brain Axis: A Comprehensive Review of Dietary Interventions in Depression and Anxiety Management

**DOI:** 10.3390/metabo14100549

**Published:** 2024-10-14

**Authors:** Mariana Merino del Portillo, Vicente Javier Clemente-Suárez, Pablo Ruisoto, Manuel Jimenez, Domingo Jesús Ramos-Campo, Ana Isabel Beltran-Velasco, Ismael Martínez-Guardado, Alejandro Rubio-Zarapuz, Eduardo Navarro-Jiménez, José Francisco Tornero-Aguilera

**Affiliations:** 1Faculty of Sports Sciences, Universidad Europea de Madrid, Tajo Street, s/n, 28670 Madrid, Spain; mariana.merino@universidadeuropea.es (M.M.d.P.); vicentejavier.clemente@universidadeuropea.es (V.J.C.-S.); alejandro.rubio@universidadeuropea.es (A.R.-Z.); josefrancisco.tornero@universidadeuropea.es (J.F.T.-A.); 2Grupo de Investigación en Cultura, Educación y Sociedad, Universidad de la Costa, Barranquilla 080002, Colombia; 3Studies Centre in Applied Combat (CESCA), 45007 Toledo, Spain; 4Department of Health Sciences, Public University of Navarre, 31006 Pamplona, Spain; pablo.ruisoto@unavarra.es; 5Departamento de Didáctica de la Educación Física y Salud, Universidad Internacional de La Rioja, 26006 Logroño, Spain; manuel.jimenez@unir.net; 6LFE Research Group, Department of Health and Human Performance, Faculty of Physical Activity and Sport Science-INEF, Universidad Politécnica de Madrid, 28040 Madrid, Spain; domingojesusramos@gmail.com; 7Department of Psychology, Faculty of Life and Natural Sciences, University of Nebrija, 28240 Madrid, Spain; 8BRABE Group, Department of Psychology, Faculty of Life and Natural Sciences, University of Nebrija, C/del Hostal, 28248 Madrid, Spain; imartinezgu@nebrija.es; 9Universidad Simon Bolivar, Barranquilla 08002, Colombia; eduardo.navarro@unisimon.edu.co

**Keywords:** depression, anxiety, nutrition, inflammation, microbiome, gut, antioxidant, diet

## Abstract

Mental health is an increasing topic of focus since more than 500 million people in the world suffer from depression and anxiety. In this multifactorial disorder, parameters such as inflammation, the state of the microbiota and, therefore, the patient’s nutrition are receiving more attention. In addition, food products are the source of many essential ingredients involved in the regulation of mental processes, including amino acids, neurotransmitters, vitamins, and others. For this reason, this narrative review was carried out with the aim of analyzing the role of nutrition in depression and anxiety disorders. To reach the review aim, a critical review was conducted utilizing both primary sources, such as scientific publications and secondary sources, such as bibliographic indexes, web pages, and databases. The search was conducted in PsychINFO, MedLine (Pubmed), Cochrane (Wiley), Embase, and CinAhl. The results show a direct relationship between what we eat and the state of our nervous system. The gut–brain axis is a complex system in which the intestinal microbiota communicates directly with our nervous system and provides it with neurotransmitters for its proper functioning. An imbalance in our microbiota due to poor nutrition will cause an inflammatory response that, if sustained over time and together with other factors, can lead to disorders such as anxiety and depression. Changes in the functions of the microbiota–gut–brain axis have been linked to several mental disorders. It is believed that the modulation of the microbiome composition may be an effective strategy for a new treatment of these disorders. Modifications in nutritional behaviors and the use of ergogenic components are presented as important non-pharmacological interventions in anxiety and depression prevention and treatment. It is desirable that the choice of nutritional and probiotic treatment in individual patients be based on the results of appropriate biochemical and microbiological tests.

## 1. Introduction

Mental health is defined as a state of well-being in which individuals can realize their abilities, cope with normal stresses, work productively and fruitfully, and contribute meaningfully to their community [1,2,3,4,5]. Disruptions in mental health typically involve abnormal thoughts and changes in perceptions, emotions, behavior, and interpersonal relationships [6,7,8,9,10]. The most common mental health disorders include depression, bipolar disorder, schizophrenia, psychoses, dementia, and developmental conditions such as autism spectrum disorders [11,12,13,14,15]. Mental illness has shown an exponential increase, particularly in Western countries. In the United States alone, 46.6 million adults reported having a mental illness in 2017, representing about 14.7% of the population. Among these, 13.3% were young people, while 7.1% were adults [16,17,18,19,20].

Preventing and managing depressive disorders has become a global public health priority due to the immense personal, healthcare, and economic burdens they impose [21,22,23,24,25]. Depression and anxiety have the highest prevalence rates, yet depressive disorders are inherently complex and multifactorial. Numerous factors such as sex, gender, socioeconomic status, social support, stress, substance use, genetic and epigenetic influences, inflammation, comorbid medical conditions, endothelial dysfunction, and diet are considered conditional risk factors for the development of these disorders [26,27,28,29,30].

New emerging hypotheses are being explored to understand psychiatric pathologies associated with nutrient intake and each person’s microbiome’s unique physical and chemical properties. The human microbiome plays a vital role in maintaining the immune system and intestinal health [31,32,33,34,35]. Nutritional intake and status are critical factors for producing neurotransmitters such as serotonin, dopamine, and norepinephrine, which regulate mood, appetite, and cognition. These neurotransmitters rely on the intake of essential nutrients like tryptophan, vitamin B6, vitamin B12, folic acid (folate), phenylalanine, tyrosine, histidine, choline, and glutamic acid. Additionally, omega-3 fatty acids play a crucial role in dopaminergic and serotonergic neurotransmission, affecting both depression and anxiety [36,37,38,39,40].

A poor-quality diet leads to inadequate nutrient intake, which may contribute to the development of mental and behavioral disorders. Thus, nutritional and ergogenic interventions are considered potential strategies for preventing these illnesses [41,42,43,44,45]. The International Society for Nutritional Psychiatry Research (ISNPR) has emphasized the necessity of incorporating nutritional interventions into psychiatric practice [46,47,48,49,50].

Moreover, the microbiome, inflammation, and epigenetics have been linked to mental health conditions. Recent studies show that certain nutrients can influence DNA methylation and histone modifications, and alterations in the gut microbiome composition can similarly affect these epigenetic markers [51,52,53,54,55,56,57,58,59,60]. The gut–brain axis is instrumental in modulating inflammatory cytokines and the production of antimicrobial peptides, which in turn affect the epigenome. This process is related to the production of short-chain fatty acids, vitamin synthesis, and nutrient absorption, as well as the intestinal production of neurotransmitters such as serotonin, dopamine, and others [61,62,63,64,65,66,67,68,69].

This review discusses the interactions between diet, the microbiome, inflammation, and mental health. However, nutritional interventions that focus on single nutrients or specific diets may not provide comprehensive benefits, as they fail to account for the organism’s multifactorial functioning. Therefore, a deeper understanding of antioxidant defense mechanisms, inflammation in various organs, and their roles in this complex system is essential.

The goal of this narrative review is to examine the role of nutrition in the development and management of depressive and anxiety disorders. The review offers guidelines and practical applications to aid healthcare professionals—including practitioners, psychologists, and psychiatrists—in their clinical practice.

## 2. Materials and Methods

To achieve the objective of this review, a comprehensive narrative approach was employed, integrating both primary and secondary sources. Primary sources included peer-reviewed scientific publications, while secondary sources comprised bibliographic indexes, websites, and databases. This narrative review focuses on examining the relationship between nutrition and mental health, with a particular emphasis on anxiety and depression. The search strategy combined specific terms to compare a broad yet relevant spectrum of studies: Nutrition OR malnutrition OR depression OR anxiety OR inflammation OR microbiome OR ergogenic OR metabolic dysfunction OR dietary intervention AND depression OR depression disorder OR anxiety OR anxiety disorders.

The search was restricted to English-language manuscripts published between 2012 and 2022, with key classical literature included when relevant. The following databases were utilized: MedLine, Cochrane, Embase, PsychINFO, and CinAhl.

The inclusion criteria were:

Studies addressing the relationship between nutrition, mental health (specifically depression and anxiety), inflammation, gut microbiome, and non-pharmacological interventions.

Articles published in journals related to nutrition, psychiatry, or psychology.

The exclusion criteria were:

(i) Research outside the specified time range, (ii) Studies not focused on the scope of the review, (iii) Unpublished studies, books, conference proceedings, abstracts, and PhD dissertations.

The search initially identified 824 articles. After removing duplicates (108 articles), 716 articles remained for screening. Following a relevance assessment based on titles and abstracts, 223 articles were removed. The remaining 493 full-text articles were thoroughly reviewed for eligibility, with 186 further excluded for not meeting the inclusion criteria. As a result, a total of 307 articles were included in the final review. Information extraction and thematic categorization were conducted by the authors, with each author focusing on areas aligned with their expertise. The resulting structure of the manuscript reflects the narrative line of the review. The primary research question guiding this review was: How do nutritional factors, particularly through the gut–brain axis and inflammation, influence the onset, prevention, and treatment of anxiety and depression?

## 3. Nutrition, Neurotransmitters and Mental Health

According to the World Health Organization (WHO), mental health can be defined as a state of individual well-being in which it is possible to carry out activities of daily living and work with acceptable performance, in addition to making a significant contribution to the development of the community to which it belongs. As mentioned above, this is achieved by using one’s own coping capacities [1,70]. However, throughout life, this state of well-being can be altered in one out of every four individuals [71], which may be associated with the development of a mental illness. Momen et al. [71] reported that certain factors such as lifestyles, daily habits and socioeconomic status can influence the development of serious medical conditions among people suffering from some type of mental illness. Indeed, it has been reported that the modification of certain lifestyle factors, such as nutrition, among others, has a direct relationship with the maintenance of good mental health [72]. Increasing evidence indicates a strong association between a poor diet and the possible development of anxiety and depression.

Moreover, mood disorders can be highly influenced by nutrient intake and energy balance. Thus, decreased appetite can be considered a main symptom of depression [73], so people with mental disorders have been reported to have worse eating habits [higher consumption of foods high in fat and sugar] than the general population [74,75]. In addition, this type of nutrient-poor diet causes the appearance of other types of associated problems, such as diabetes or cardiovascular events [76]. However, an interesting study [77] reported that although depression is associated with poor dietary habits, these may improve in the long term in people suffering from this type of disease, rejecting the hypothesis of reverse causality. Considering the different stages of life, from preconception to old age, the role of nutrition in preventing mental disorders has been well documented [78]. It is known that several studies have reported that maintaining a good nutritional status during childhood and adolescence prevents the risk of suffering from some type of mental illness during these stages [78,79]. However, it is more common to observe mental diseases [e.g., symptoms of depression] [80] in the later stages of life, which are also often accompanied by other mental problems. Gregorevic et al. [80] reported the important role of diet in preventing the development of Alzheimer’s and vascular dementia.

The dietary profile could play a key role in the preservation of cognitive function [81]. An adequate intake of micronutrients and macronutrients through a balanced diet has been shown to have beneficial effects on brain function [82,83]. However, the absence of this type of element, especially minerals, causes dysfunctions at the cognitive level [84]. In this sense, it has been shown that a deficient intake of vitamin B12 [85] and B9 [85] can increase the risk of developing depression during adulthood. Thus, Kwok et al. reported that B12 supplementation for 24 months in elderly people produced a significant reduction in symptoms of depression. Nevertheless, Okereke et al. [86] conducted long-term folate and B12 supplementation in healthy women. Their results showed that such supplementation does not reduce the risk of developing depression in this population. In addition, B12 supplementation did not increase cognitive function or decrease the risk of dementia in older adults [87]. These conflicting results may be due to the different amounts of vitamins used [88]. In relation to vitamin D, its deficiency can affect different brain neurotransmitters, causing different mood disorders [89]. In this regard, Desrumaux et al. [90] observed in murine mice that a deficiency in vitamin D is associated with increased anxiety in adulthood. In human models, Annweiler et al. [91] reported that a person with depression would have a 65% chance of having a deficient concentration of 25-hydroxyvitamin D than a healthy person. Accordingly, a meta-analysis has shown that a positive association between low vitamin D levels and depression existed [92].

Moreover, different nutritional psychiatry trials have shown that deficient levels of essential trace elements can have a negative effect on mental health [93]. In this respect, deficient magnesium levels raise cortisol concentration, increasing the risk of neurodegeneration [94]. In animal models, a deficiency in dietary magnesium levels has been linked to the development of anxiety [95]. In relation to selenium, Casaril et al. [95] reported the beneficial function of this element in ameliorating depression related to oxidative stress and inflammation in mice [95,96].

Theoretical reasoning established that the pathophysiology of anxiety and depression can be explained by some neurotransmitters such as serotonin, dopamine, and noradrenaline [97]. Serotonin has been established as one of the most important neurotransmitters influencing mental health [98], and its alteration may be related to the development of mental diseases [99]. Moreover, people experiencing traumatic stress have been reported to have low serotonin and high dopamine levels, which can lead to psychotic behaviors [100]. Indeed, low dopamine levels are correlated with the development of major depressive disorders [101]. However, despite the above, contemporary neuroscience research has failed to confirm that mental disorders can be fully explained by neurotransmitter deficiency alone [102]. In addition to the neurotransmitter hypotheses, the etiology of depression also involves hypothalamic–pituitary–adrenal (HPA) axis dysregulation, neurogenesis, and synaptic plasticity. Dysregulation of the HPA axis has been linked to the body’s stress response, contributing to the development of depression by altering cortisol levels and inflammatory markers.

Moreover, neurogenesis, especially in the hippocampus, is believed to be impaired in depressive disorders, leading to reduced cognitive function and emotional regulation. Similarly, changes in synaptic plasticity, which refers to the brain’s ability to adapt and reorganize neural connections, are thought to play a crucial role in mood disorders. These factors highlight the multifaceted nature of depression, where neurotransmitter imbalances are just one piece of a more complex puzzle that includes neuroendocrine function and brain plasticity [102]. By integrating these additional elements, it becomes clear that depression’s pathophysiology is not solely neurotransmitter-dependent but also involves broader neurobiological processes.

## 4. Gut, Inflammation, and Mental Health

The human body has a large collection of microorganisms, including bacteria, viruses, protozoa, fungi, and archaea. These form the so-called microbiome. Specifically, we focus on the gut microbiome, gut flora, or microbiome, which is defined as the microorganisms that live in the digestive tract. Generally, bacteria have been associated with disease; however, most of the microbiomes are related to health and metabolism. Considering that there are more bacteria than cells in the human body [40] trillion bacterial cells vs. only 30 trillion human cells], these may weigh roughly as much as the brain, functioning as an extraordinary organ [103]. Indeed, the gut microbiome collective genome exceeds over 100 times the amount of human DNA in the body; this represents an enormous genetic potential and a vital role in virtually all physiological processes in the human body.

The role of gut microbiota in the brain and mental health is increasingly attracting the attention of neuroscientists and psychiatrists [104]. The first paper about the potential relationship between the gut and the brain was published just a decade ago [105] after being rejected seven times. Nowadays, the gut–brain axis is being featured at major conferences discussing neuroscience since it might be easier to manipulate the gut than the brain [106]. The gut–brain axis refers to the bi-directional pathway between the gastrointestinal system and the brain, more specifically, between the gut microbiome (the largest population of microorganisms in the human body) and the brain [107,108]. Moreover, gut bacteria have been linked to several mental illnesses, and patients with various psychiatric disorders such as depression, bipolar disorder, schizophrenia, and autism have been found to have significant alterations in the composition of their gut microorganisms. These alterations can be modifiable through intervention pathways, such as nutritional and ergogenic strategies or behavioral factors, such as physical exercise, highlighting the importance of remodeling intestinal microbiota.

However, there are several elements that have an impact on the conformation and composition of the microbiota, among them diet, physical exercise, stress, medication, geography, lifecycle stage, mode of birth, and infant feeding method. Regarding the origin and conformation of the microbiota, the microbiome of vaginally delivered infants is like the maternal vaginal microbiome, and that of infants delivered by cesarean section resembles the maternal skin microbiome [109]. Another factor of early development is the mode of feeding since studies suggest that those who are breastfed have a richer microbial environment and a more protective profile against the development of allergies or diseases at later ages [110]. However, another of the major determinants of gut microbiota composition is carried out in the phase of adulthood, mainly diet and physical exercise [111]. Furthermore, exposure to stress is believed to be the third most important factor after diet and exercise to alter gut microbiome composition [112]. However, the microbiota is a living, changing, adaptable entity given epigenetic and behavioral conditions. Therefore, regardless of other conditioning factors such as the mode of delivery and infant feeding method, environmental conditions, medications, stage and mode of lifecycle, comorbid diseases, and medical procedures, it can be modified to benefit the patient [113].

A disruption to the microbiota homeostasis caused by an imbalance in their functional composition and metabolic activities or a shift in their local distribution is termed dysbiosis, indicating microbial imbalance or maladaptation [114]. Thus, dysbiosis will produce alterations in the bidirectional communication between the gut and brain, the so-called the gut–brain axis. Therefore, alterations in the central nervous system through neural, endocrine, and immune routes, thereby controlling brain function, are expected. Specifically, there are four main ways in which gut-resident bacteria can influence neurons and the brain: blood vessels, neuropod cells, enteroendocrine cells, and immune cells [115]. The production and synthesis of certain metabolites depends on the population of bacteria, eukaryotes, archaea, and viruses within the microbiota (Figure 1). Likewise, the composition of the microbiota tends to change during the life span, and this is in turn modulated by contextual, behavioral, genetic, and epigenetic factors, where two factors have received recent attention: physical activity and nutrition [116].

Some research supports the essential and substantial role of gut microbiome in the regulation of anxiety, mood, cognition, and pain. Authors suggest that the utilization of strategies and tools that modulate the microbiome composition may be an effective strategy for developing novel therapeutics for central nervous system disorders. In reference to mental health, research has accelerated, especially in the last two years, with the advent of improved molecular and metagenomic tools. In a recent meta-analysis, the authors suggest that there is sufficient scientific evidence to support the relevance of the bidirectional gut microbiota–brain communication in mood disorders in humans, such as the effect of probiotics on brain connectivity and mental health outcomes and pregnancy-related stress on gut microbiota in the newborn child [117]. Yet, most of the analyzed studies are in their early stage and these studies differ in terms of study design, study populations, microbiota analyses, and outcomes.

In addition, recent studies have found that the vagus nerve plays a central role in the crosstalk between gut microbiota and the brain [118,119,120,121]. This finding makes sense since the vagus nerve is the longest cranial nerve from the brain to the rest of the body, including the colon and from the gut bacteria back to the brain [106]. This would explain why the gut is sensitive to emotions, and some experiences can make us feel sensations described as “gut-wrenching”, nausea, “butterflies”, or “stressed out”. In addition, why the brain is sensitive to gastrointestinal function, and therefore, nutrition may play a significant role in emotions and mental health.

In addition to the vagus nerve, the gut microbiota influences the brain via the transmission of signaling molecules through the circulatory system and across the blood–brain barrier, also affecting the immune system [120]. In fact, the immune system is a key pathway between the gut and brain with an important role in stress-related psychiatric disorders, since disruption of the gut–brain axis may lead to chronic inflammation and hyperactivity of the hypothalamic–pituitary–adrenal axis [104,118,122,123,124,125]. Moreover, the brain shapes the gut microbiota mainly through the stress-mediated neuroendocrine system [126,127].

The gut–brain axis involves interactions between the stress system and the immune response [128,129]. This relation is particularly evident in gastrointestinal disorders (also considered stress-related conditions) such as irritable bowel syndrome, an idiopathic inflammatory condition of the gastrointestinal tract whose natural history is one of periods of remission and relapse exacerbated by pro-inflammatory effects of stress, or others such as Crohn’s disease or ulcerative colitis [128,130,131,132]. Basically, chronic stress exacerbates intestinal inflammation, dysregulating the gut–brain axis, which is involved in those gastrointestinal diseases [133].

In the last 15 years, there has been a growing appreciation of the role of gut microbiota in all aspects of health and disease, including mental health, although more research is needed [124]. Indeed, microbiota dysregulation or dysfunction within the gut–brain axis [dysbiosis] has been documented [134] and has been related to anxiety and depressive-like behaviors [107] and even may increase the perception of pain since both the gastrointestinal tract and the brain contain opioid receptors [135,136]. Accumulating evidence suggests that dysfunction of the gut microbiota contributes to the pathophysiology of a wide range of neuropsychiatric disorders [107,119,125,126,127,137,138,139,140,141,142,143]. In particular, the most extensively studied conditions are anxiety, depression, and schizophrenia.

Dysregulation of gut microbiota [and chronic stress] leads to hypothalamic–pituitary–adrenal axis dysregulation involved in exaggerated response to stress and inflammation associated with increased risk of anxiety and depressive symptomatology [107,137,139,144,145,146]. Interestingly, some metabolites released by various microbes [for example, short-chain fatty acids] can attenuate the hypothalamic–pituitary–adrenal axis and vagal nerve responses involved in anxiety and depression [147]. Dinan and Cryan referred to the link between the gut and depression as “melancholic microbiota” [148]. In turn, chronic stress and depression can also significantly impact the gut–brain axis at all stages of life [108,126,146], thus altering the microbial profile in the colon [149]. Also, dysregulation of the microbiota of the gut–brain axis also seems to increase the risk of schizophrenia [147,150], again by inducing dysregulation of hypothalamic–pituitary–adrenal axis and inflammation [147]. Moreover, new research suggests that gut microbiota can also affect drug absorption and metabolism of psychoactive drugs used for treating such conditions.

Furthermore, the gut–brain axis plays a major role in the pathogenesis of many neurodegenerative disorders, including Alzheimer’s disease [106,118,151], Parkinson’s disease [107,113,137,142], amyotrophic lateral sclerosis, multiple sclerosis, and Huntington’s disease (HD) [152].

Aging, associated with a narrowing in the diversity of the gut microbiota and exposure to chronic inflammation [118], together with poor diet and dysbiosis or dysregulation of gut microbiota, for example, can result in a poor diet gut-derived inflammatory response due to dysbiosis, and may promote neuroinflammation involved in the pathogenesis and progression of Alzheimer’s disease, the most common cause of dementia [153,154,155,156,157,158,159]. Indeed, recent research suggests that an impaired gut microbiome can trigger neuroinflammation and cerebrovascular degeneration, inhibit the autophagy-mediated immune system, and affect the brain through the vagal afferent fibers [160].

In sum, the gut–brain seems to be involved in the pathogenesis of most neuropsychiatric disorders and neurodegenerative conditions that may involve brain inflammation [125]. This is important because even low-grade chronic inflammation plays an important role in most mental health problems, ranging from anxiety and depressive symptoms to dementia, and gut inflammation has effects on the brain [161].

Finally, recent findings on the gut–brain axis provide an opportunity to develop novel therapeutic approaches for the prevention and treatment of mental health problems by targeting and modifying the healthy gut microbiota. Indeed, since gut microbiota is susceptible to stress, microbiota-targeted interventions might be used to improve highly prevalent stress-related neuropsychiatric or neurodegenerative conditions in the near future, such as gastrointestinal diseases or anxiety and depression, among others [126].

In particular, diet is one of the most critical modifiable factors of the gut microbiota ecosystem, for example, using probiotics [81,158,162]. Indeed, targeting the gut microbiota through prebiotic, probiotic, or dietary interventions to modulate the gut–brain connection may be an effective “psychobiotic” strategy for treating neurodegenerative and neurodevelopmental disorders [120,134,148,154,158,163,164,165,166,167]. In other words, a psychobiotic is a live organism, a class of probiotics that, when ingested in adequate amounts, produces a health benefit in patients with mental health problems by rearrangement of the intestinal microbiota [168]. Moreover, those benefits are related to their anti-inflammatory effects and hypothalamic–pituitary–adrenal axis activity reduction. However, further studies are required [148,169,170,171].

These studies are important because they invite us to reconsider the traditional distinction between mental and physical health and support a more integral view of health, where mental health might benefit from anti-inflammatory nutrition/diet and stress-reduction-based psychological interventions. Future research should explore further the markers of the gut–brain interactions, including serum/salivary cortisol (as a marker of the hypothalamic–pituitary–adrenal axis), heart rate variability (as a marker of the sympathovagal balance), or brain imaging studies.

## 5. Nutrition Behaviors and Mental Health

Current knowledge about nutritional behaviors and mental illness focuses on prevention and the study of underlying triggers. Today, more than 500 million people globally suffer from depression and anxiety, making it a significant public health challenge [18]. Evolutionarily, acute stress can promote adaptation, but chronic stress often leads to pathophysiological processes, such as impaired immune function, metabolic disturbances, and inflammation [19]. The modern Western lifestyle, characterized by high caloric intake and low-quality diets, has been implicated in increasing susceptibility to physical and mental health disorders, including cognitive decline, hippocampal atrophy, and psychiatric illnesses. This is particularly exacerbated in environments that promote chronic stress [20,172,173,174].

Both depression and anxiety have been linked to an increased allostatic load, which refers to the wear and tear on the body due to chronic stress [19,21,175]. Several biomarkers related to this adaptive process have been associated with a higher likelihood of developing mental disorders [22,176,177]. These biomarkers are influenced by various factors, including work-related stress [23,24,178,179,180], early adverse life events [25,26,27], and unhealthy dietary patterns such as high caloric intake and obesity [28,29,30,181].

Animal models have shown that inadequate intake of certain amino acids can induce anxiety-like behaviors [31,182,183]. In humans, persistent or intense state anxiety has been associated with physiological imbalances [32]. Factors such as social competitiveness, work environment, and individual coping mechanisms play essential roles in how anxiety manifests [33,34,35,184]. Some research suggests that metabolic improvements achieved through healthier eating habits may offer protection against anxiety disorders [4,36,37,38,185,186,187]. Clinical trials using small sample sizes have observed reductions in self-reported anxiety when comparing control groups with those assigned to intervention programs combining physical activity and healthy diets [39,40,188]. However, there is still no scientific consensus on the statistical robustness of theories linking dietary improvements to mental health [41,42]. While some studies report reduced stress following dietary supplement interventions [43,189], the evidence remains insufficient to conclude that supplementation with specific nutrients or alternative treatments provides benefits beyond the placebo effect [44,190]. This may be because anxiety and depression are multifactorial disorders, limiting the effectiveness of single interventions or preventive strategies. Variations in age, gender, socioeconomic status, and brain structure further complicate treatment approaches [45,46,47].

Both animal and human studies show that stressful life events, natural disasters, and psychiatric disorders like anxiety and depression can suppress appetite. Conversely, diets high in saturated fats and refined sugars can contribute to maladaptive stress responses and exacerbate mental health conditions [48]. Many eating disorders are closely linked to the socio-emotional functions of the brain [49,191]. Anxiety can act as a modulator of pathological processes, particularly during adolescence and school-age years [50,51,52], but also in adults, with a higher prevalence among women and individuals from disadvantaged socioeconomic backgrounds [53,54,192,193]. Poor quality of life and psychological well-being are associated with a polarization of psychiatric disorders, such as anorexia nervosa [55,194], obsessive–compulsive disorder [56], negative emotions [57,58], and other metabolic disturbances [42,195]. These disorders may result from chronic or acute allostatic load, impacting both organ systems and brain structures [59]. Consequently, individuals with anxiety and depression, particularly those with severe or chronic symptoms, often have poorer diets and restricted caloric intake [60,61].

Dietary changes have been shown to affect gut health, which plays a role in some psychiatric disorders [62,63]. This connection between brain function, physiology, and eating behaviors highlights the importance of diet in mental health [64,65]. People with anxiety and depression tend to have less diverse gut microbiota, which is part of a complex, bidirectional communication network coordinated by the neuroendocrine, immune, and enteric systems. Intestinal peptides act as key regulators in this system [66,196]. Dietary patterns have been directly linked to anxiety and depression, with healthier diets—comprising fruits, vegetables, legumes, and essential fatty acids—reducing the risk of these conditions. In contrast, Western diets rich in red meat, saturated fats, refined grains, and processed foods but low in vegetables disrupt gut health, increase inflammation, and worsen mental health outcomes [67,197]. For instance, adherence to the Mediterranean diet has been associated with positive affective states, improved physical fitness, reduced anxiety, decreased depressive symptoms, and lower perceived stress [68,69].

In summary, there is substantial scientific evidence supporting the relationship between nutritional behaviors and mental disorders such as anxiety and depression. Mediators of the physiological and psychological adaptation processes, which contribute to the allostatic load, make it more likely to observe dietary dysfunctions in both animal models and humans. Healthy diets rich in vegetables and essential fatty acids show promise in intervention and prevention programs for anxiety, depression, and perceived stress, as these diets are closely linked to a healthy gut and better overall mental health.

## 6. Nutritional Interventions in Depression and Anxiety

Many scientists and the WHO recognize that a balanced diet is essential for good physical health. However, there is greater controversy in the scientific community regarding the role of nutrition in the onset and progression of mental illnesses and behavioral problems, and the role of diet in therapeutic efficacy in patients and the progression of disorders is still unclear. Various psychopathological states. In this sense, there does seem to be a consensus regarding the identification of nutritional styles and the prevalence of mood disorders. Thus, it can be verified that there is a correlation between patients with depressive symptomatology and the absence or low intake of micronutrients, polyunsaturated fatty acids, and certain proteins.

Similarly, a poor intake of minerals such as manganese, zinc, and copper, as well as a low intake of fatty acids, is associated with the presence of high levels of anxiety. It is important to consider that nutritional intervention should not be based on a unique dietary aspect but should extend the study to the individual functioning of each patient. This is consistent with specific genetic and metabolic determinants, which should be studied to implement a multifactorial evaluation. Similarly, other factors, such as the quality of basic functions like sleep, are associated with certain nutrients such as vitamin C, calcium, or selenium intake. It is known that well-functioning sleep directly impacts the state of the organism and its way of processing the nutrients that are ingested. Although it is evident that the origin and maintenance of mental pathologies will be multifactorial, it seems logical to conclude that exercise, sleep and diet are the backbones that allow their prolonged maintenance over time [18]. In this sense, the enteric nervous system performs an important task in regulating the digestive system. This cellular structure is part of the autonomic nervous system, and its function is to control intestinal functions such as inflammation of the organs of the digestive system, motility, and secretion. In other words, it regulates food intake, metabolism, and digestion. This is the most complex area of the autonomic nervous system and is made up of a high concentration of neuronal and glial cells. In addition to this high specialization, it also has the function of controlling the release of serotonin. Therefore, alterations in the functioning of this system will have a negative impact on the organism and will favor the appearance of psychiatric disorders.

In the following sections, we review current studies examining the role of specific dietary components in interventions targeting anxiety and depression among adults, children, and adolescents.

There is a persistent relationship between anxiety and depression and severe adverse effects on cellular, neuroendocrine, cardiovascular, and central nervous system immunity [198]. Anxiety and other depression-like mental illnesses increase oxidative stress, increase inflammatory markers and overactive stress pathways, altering neuronal synapses and producing other structural brain changes [4]. A meta-analysis of randomized controlled trials of the effects of dietary improvement on symptoms of depression and anxiety found that dietary interventions significantly reduced depressive symptoms. However, no effect of dietary interventions for anxiety was observed. However, studies that only included women found significantly greater benefits from dietary interventions for both depressive and anxiety symptoms [42]. In a systematic review, Aucoin et al. found that less severe anxiety states were related to a diet consisting of caloric restriction, breakfast consumption, supplements of a broad spectrum of micronutrients-macronutrients, probiotic consumption, and intake of a variety of fruits and vegetables.

In this line, it is known that some herbs and spices are traditionally used to cook food and have great health benefits [199]. Some facilitate the reduction of fats, salts and sugars and thus increase the antioxidant capacity. Turmeric, ginger, and garlic are the ones that have obtained the best results and are the most reliable in prevention. Turmeric acts at the cellular level and reduces the toxicity of the organism through the stimulation of the lymphatic system. In the same way, it acts as a protector against oxidative stress in neurons [200]. Because of its antioxidative and anti-inflammatory properties in the brain, curcumin, which is the phytochemical that gives turmeric root its color, can be used as an alternative to chemical treatment in mood disorders. Saffron has also shown some efficacy in improving the reconciliation of sleep, and a daily dose of about 30 mg significantly improves depressive symptomatology [201]. It has been shown to have similar effects to fluoxetine or Imipramine [202]. This is explained by its antioxidant effect on neuronal degeneration in the central nervous system. Likewise, integral cereals and fruits are related to a tendency to increase the level of alertness and a reduction in stress [203]. Studies in this line indicate that the consumption of cereals and certain fruits has an impact on the reduction of depressive and anxiogenic symptoms [204]. This is explained by the fact that cereals have an anti-inflammatory action, reducing blood pressure and associated pathologies, both physical and physiological. Specifically, C-reactive protein, produced by the liver, is a marker used in the study of cardiovascular pathologies, and the measurement of the concentration of this protein is used for the evaluation and prevention of different pathologies [205]. Therefore, an intake that includes the consumption of cereal fiber is associated with less inflammation of the organs and an improvement of intestinal functioning [206].

On the other hand, the most severe states of anxiety were associated with a diet high in fat, highly refined sugars, cholesterol, trans fats, and tryptophan [207]. On the other hand, fasting interventions have previously been shown to be qualitatively effective in relieving stress, anxiety, and depressive symptoms. Berthelot et al., in an integrative study of clinical trials with a low risk of bias, found that fasting groups had statistically significantly lower levels of anxiety, depression and body mass index compared to controls without increased fatigue. However, the authors report that the results are preliminary and inconclusive but encouraging and suggest that fasting appears to be a safe intervention [208]. The above findings are consistent with the established evidence on healthy eating patterns and increases in mental health and the benefit of complementing classic treatments such as medication and psychological therapy with dietary interventions according to each patient’s specific needs. The low cost and high effectiveness of these supplemental plans may also confer additional benefits to physical aspects of health.

Recent advances in metabolomics offer a promising approach to studying the complex interactions between diet, microbiome, and mental health, including anxiety and depression [181]. Metabolomics allows for a comprehensive analysis of metabolites in biological systems, providing insight into the metabolic changes that occur in response to dietary patterns and their potential impact on mental health [185,186,187]. Although few studies have applied metabolomics specifically to this area, the potential of this technology lies in its ability to capture the dynamic interplay between nutrients, the gut microbiome, and neurochemical processes [42]. Future research could leverage metabolomics to uncover biomarkers related to mental health disorders and dietary interventions, offering a holistic and personalized approach to managing anxiety and depression through nutrition.

## 7. The Role of Antioxidant Defense and the Effect of Dietary Interventions for Depression and Anxiety

The antioxidant defense system prevents oxidative cell damage and erythrocyte oxidation in circulation [209]. Under normal physiological conditions, reactive oxygen species (ROS) play an essential role in the activation of some transcriptional factors and cellular signaling and in the regulation of vital physiological processes (i.e., phagocytosis, apoptosis, fertilization, and activation of certain transcriptional factors) [210]. Oxidative phosphorylation produces the major amount of ATP. This process happens in the mitochondria of the cell and promotes ROS free radicals, among others (reactive nitrogen species, carbon and sulfur-centered) [211,212]. Regarding the nervous system, moderate levels of ROS have a key role in the development and growth of neurons [212].

Under normal conditions, cells contain sufficient enzymes (i.e., superoxide dismutase, catalase, and glutathione peroxidase) to protect against free radical injury [193]. However, when an imbalance between the production of ROS and the antioxidant capacity of the cell occurs, oxidative stress (OS) will occur [213]. The OS generates a signal of pro-inflammation and macromolecular damage and triggers cellular apoptosis [212]. Therefore, OS is intimately related to inflammation, which activates the immune system to defend against such threats, with the aim of minimizing tissue damage and preventing systemic spread [214]. In the context of psychological diseases, there is evidence for a chronic, low-grade activation of inflammation and the immune system. It is considered a decline associated with chronological age and appears with the passage of time. However, external factors also cause imbalances in the immune system that allow this process to develop more rapidly. However, the organism has designed antioxidant defense mechanisms to protect and repair the mitochondria, enzymatic antioxidants such as superoxide dismutase and non-enzymatic antioxidants such as vitamin E, vitamin C, selenium, or coenzyme Q-10. In this respect, the diet provided to the organism can be designed with antioxidant supplements that will allow a better way to fight oxidative stress and delay cell aging [212].

In this way, the higher lipid content, higher oxygen consumption and the lower capacity of the antioxidant defense system of the brain are related to the higher OS associated with this organ [212]. It has been argued that OS is a potential neurobiological contributor to the development of psychological diseases [i.e., schizophrenia, obsessive–compulsive disorder, psychosis, or depression] [212], and it is considered a possible contributor to their pathophysiology.

Therefore, proinflammatory signaling and OS are considered key factors in the pathogenesis of psychological diseases [212], and they have opened a new target for the treatments of these pathologies. In this way, antioxidant and uninflammatory drugs are a new potential treatment target. Products such as N-acetylcysteine [215] and polyunsaturated fatty acids [216] represent an interesting therapy. In addition, there is emerging evidence showing a strong influence of diet and gut microbiota on neurological processes and the physiopathology of some psychological diseases [217].

Regarding dietary interventions in depression, non-pharmacological approaches to the treatment of this pathology are gaining a reputation. Incipient evidence supports the importance of lifestyle factors such as diet, smoking cessation or physical activity in depression development [41]. A previous study found a small to very large effect of dietary intervention on depression scores, which is like that obtained after applying pharmacotherapy or psychotherapy [218]. Moreover, studies have also found that the benefits of incorporating certain nutrients such as magnesium, zinc, and other vitamins into the diet are directly related to significantly improving mood balance. These studies present healthy behavioral patterns and the prolonged intake over time of foods with antioxidant capacities and the maintenance of normative values of fatty acids, cholesterol, proteins, vitamins, and minerals necessary for the proper functioning of the body’s systems.

From an epidemiological point of view, previous studies reported a relationship between healthy dietary patterns and a lower incidence of depression [219]. These healthy dietary patterns are characterized by the abundance ingestion of fruits, vegetables, fish, nuts, cereals, seeds, legumes, olive oil, and the moderate dairy incorporation of unsaturated fats and eggs [220]. Accordingly, the Mediterranean, Norwegian, and Japanese diets include the same dietary patterns [70]. In contrast, a low-quality diet, consisting of high consumption of processed foods, refined grains, red meat, high-fat, fried, and sugary foods, and low fruit and vegetable ingestion, is associated with a greater incidence of depression [221]. Interestingly, a concomitant increase in depressive disorders and a lower healthy lifestyle behavior (e.g., lower diet quality) has been found during the last decades [4]. Therefore, diet is a modifiable factor that is associated with less depressive symptoms [4], and specifically, some healthy diets, such as the Mediterranean diet, promote an anti-inflammatory effect that can affect the risk of depression disorder [213].

According to a previous meta-analysis [222], diets with different macronutrient compositions do not affect depressive symptoms, although they are improved by weight-loss diet interventions. However, a relationship between macronutrients and depression has been reported in epidemiologic studies [223,224], which found a lower risk of depression if the diet includes a higher intake of fish, fiber and omega-3 fatty acids and a positive association between higher consumption of sugar and refined carbohydrates and depression. Actually, more clinical research that analyzes the effect of macronutrient composition diet on depressive symptoms in people with depression because the evidence reported using non-depressed individuals, and perhaps, for this reason, these studies found a limited effect of varying fat, protein, and carbohydrate intake on depressive symptoms.

On the other hand, micronutrients are involved in metabolic pathways that affect the development and optimal functioning of the nervous system. Thus, poor ingestion of micronutrients affects the burden of depression symptoms. They can modify the production and activity of neurotransmitters such as serotonin, dopamine, and norepinephrine, which are involved in the regulation of mood, appetite, and cognition [225,226]. In addition, they can modify the oxidative stress and inflammatory processes and the hypothalamic–pituitary–adrenal system and glutamatergic signaling [227]. In this way, the micronutrients associated with depression burden include zinc, iron, magnesium, selenium and B vitamins (B6, B12), folic acid, vitamin D, tryptophan, phenylalanine, tyrosine, histidine, choline, and glutamic acid [4,227,228,229,230,231,232]. Therefore, from a practical application point of view, a diet with a high proportion of vegetables and fruits is recommended due to the higher amounts of these micronutrients that these types of food contain. Additionally, the antioxidant effect of plants must be taken into consideration because they also contain effective phytochemicals (e.g., vitamin C, polyphenols, or flavonoids), which can have antidepressant effects. It is known that foods that have a vegetable origin apply significant health benefits and are associated with a clear decrease in the risk associated with the appearance of certain pathologies ranging from cardiovascular diseases to mood disorders. This is possible due to the mechanisms of preventive action that appear with its consumption, such as modifications in metabolism, lower blood pressure and antioxidant action, among others. It is precisely this last effect that works as a neuroprotector that allows the CNS to defend itself from neurological damage related to degenerative diseases and the inflammatory effect related to depression and anxiety. [233,234,235,236].

In the same way, anxiety disorder may be affected by dietary patterns and nutrition. However, some controversial results are reported in the literature because a previous meta-analysis showed no effect of dietary interventions on anxiety [42]. Still, another review study demonstrated that nutrition reduced anxiety [4]. As we explained before, the metabolism of neurotransmitters like dopamine or noradrenaline [4] can be affected by some nutrients (C and B vitamins, zinc, magnesium, etc.) but also by other factors such as chronic stress, increasing the risk of anxiety by the lower synthesis of neurotransmitters [237] and the neuronal membrane structure [238].

In this way, diet patterns are associated with the risk of anxiety. Evidence showed that adherence to healthy dietary patterns such as the traditional Chinese diet [gruel, oatmeal, whole grains, fresh yellow or red vegetables, fruit, and soy milk] [239], the Mediterranean diet or traditional Australian diet [fruit, vegetables, meat, fish, and whole grains] [240] had been associated with a lower risk of anxiety disorders. Moreover, the odds of anxiety are also associated with lower healthy diet scores in a sample of women from Norway [241]. Furthermore, adherence to other diet patterns, such as the vegan diet [242], lacto-vegetarian diet, and a diet that includes daily vegetables and fruit intake [243] reduced the risk of anxiety. Therefore, based on epidemiological studies, higher diet quality (characterized by low sugars and a high number of vegetables, fruits, nuts, legumes, grains, and lean protein) is associated with reduced risk of anxiety, while higher odds of anxiety are associated to dietary patterns that do not meet food-based recommendations. Remarkably, some micronutrients affect anxiety disorders. There is some evidence that amino acids (lysine and arginine), minerals such as zinc or magnesium and some vitamins (B vitamins, vitamin C, and vitamin E) can be used as adjuvant therapy in the treatment and prevention of anxiety [4]. Therefore, a healthy dietary pattern that meets macro and micronutrient recommendations may favorably affect anxiety. Thus, following current evidence, healthy dietary patterns to have benefits for anxiety should include vegetables, fruits, whole grains, low-fat dairy, lean protein foods, fatty fish high in n-3 fatty acids, and olive oil [4]. Finally, although vegan and vegetarian diets are classified as Healthy Eating Patterns, these types of diets should warrant the correct intake of some nutrients like iron, B12 vitamin and n-3 fatty acids to reduce the risk of anxiety and depression disorders [4].

Interestingly, a new therapy focused on anti-inflammatory and antioxidant products to treat anxiety and depression has emerged during the last few years. Specifically, N-acetylcysteine has demonstrated beneficial effects on different psychiatric disorders (including posttraumatic stress disorder, bipolar disorder, attention deficit hyperactivity disorder, anxiety, depression, obsessive–compulsive disorder, obsessive–compulsive-related disorder, and schizophrenia) because it affects the regulation of several neurotransmitters, inflammatory mediators, and oxidative homeostasis [244]. Although N-acetylcysteine preclinical studies have shown a positive effect to ameliorate depressive [245] symptoms and improve functionality, the clinical efficacy of this pharmacological therapy is controversial [246]. In the same way, it has been analyzed that the effect of non-steroidal anti-inflammatory drugs on depression and its efficacy on depressive symptoms appears to be negligible [246].

Another adjunctive therapy for depressive and anxiety symptoms that some expert associations have included in their practice guidelines for the treatment of depressive patients is omega-3 fatty acids [247]. In this way, poly-unsaturated fatty acids (PUFA), mainly docosahexaenoic acid (DHA), eicosatetraenoic acid (EPA), and arachidonic acid (AA), as well as their ratio of, affect neural function, decreasing cell membrane permeability and also increasing systemic and brain inflammation if the ratio of DHA to AA decrease [248]. In addition, n-3 fatty acids regulate dopaminergic and serotonergic neurotransmission, modifying depressive burden [13]. For these reasons, PUFA supplementation has been included as adjunctive therapy, showing a beneficial effect of adding this supplement to treatment in people with depression [249]. Still, it is not recommended for depression prevention [250]. In this last case, if the target is depression prevention, a healthy dietary pattern with food that contains a high amount of n-3 fatty acids should be taken [4]. Furthermore, current evidence suggests that n-3 fatty acids benefit anxiety disorders [4].

In summary, healthy dietary patterns characterized by vegetables, fruits, olive oil, whole grains, nuts, lean protein sources, fish and seafood are important to prevent and manage depressive and anxiety symptoms and to promote optimal mental health. In addition, healthy dietary patterns should fulfill the recommendations of magnesium, zinc, vitamin B12 and n-3 PUFA to prevent the risk and depression and to ensure normal psycho-physiological function. Some specific populations, such as vegans, should pay more attention to fulfilling these diet recommendations to avoid any nutrient deficiency. More research is needed to increase the knowledge of the effect of nutrition and diet, their mechanism on psychological disorders, and their impact on the patient’s health and social context.

## 8. Ergogenic Interventions in Anxiety and Depression

There is significant evidence of the importance of nutrition in promoting mental health and how improvements can be implemented using ergogenic interventions [251,252]. Nutritional ergogenic aids refer to all those nutritional supplements that serve as performance enhancement aids. In short, there are physiological, mechanical, psychological, pharmacological, and nutritional ergogenic aids. On the other hand, to understand the relationship with mood disorders, it is important to understand the biological basis for them. Anxiety is a psychophysiological response that is triggered by a stimulus considered to be aversive, real or [253] imagined. When there is no real relationship between the response and the anxiogenic stimulus, it is referred to as a disorder. Anxiety produces specific motor and endocrine responses. The motor response is modulated by the thalamic pathway, the limbic system, and the amygdala [254]. It is a rapid and direct response that allows information to be obtained and the appropriate response to the perceived threat to be sent. It is the so-called fight or flight response [255]. On the endocrine side, this is the activation of the hypothalamic–pituitary–adrenal axis, releasing cortisol into the prefrontal regions of the brain, which implies a difficulty in neuronal synapses in the hippocampus. In addition, serotonin is also involved in this response [253]. As for depression, the main neurobiological substrate of this disorder is an alteration in the neurotransmitters’ serotonin and noradrenaline [256]. This alteration also causes endocrine alterations in the secretion of thyroid hormone and cortisol [257,258]. In both cases, serotonin is a fundamentally affected neurotransmitter [259,260]. Also known as 5-hydroxytryptamine, it is caused by metabolizing a basic amino acid such as tryptophan [261,262]. In this case, the enzyme tryptophan hydroxylase modifies tryptophan and metabolizes it into 5-hydroxyltryptophan (5-HTP), which is converted into serotonin by decarboxylation [81,263,264].

It was found that the deficiency of tryptophan was related to higher levels of anxiety fact, which could be related to the comorbidity with Binge Eating Disorder in people with anxiety and depression [265,266]. This is because people with mood and emotional disorders seek out food in order to reduce anxiety [267]. This amino acid releases serotonin, and it is known that a serotonin deficit will facilitate the appearance of symptoms such as sadness, anxiety and irritability [268]. In this line, poor nutrition can lead to a resistance to antidepressant pharmacology of up to 40% [269,270]. For these cases, it has been shown in recent years that increasing the intake of folic acid or n-3 polyunsaturated fatty acid facilitates better results in these treatments for mood disorders [81]. Recent studies have shown that an unbalanced diet is directly associated with a greater tendency to develop mental illnesses such as bipolar disorder, obsessive compulsive disorder [OCD], depression and anxiety, among others [271,272]. Folic acid has been shown to be an adjuvant in antidepressant treatment and is therefore administered in conjunction with pharmacology such as serotonin reuptake inhibitors [273]. A dose of 10 mg daily has been shown to reduce plasma homocysteine and increase the efficacy of the medication with which it is administered [274]. Recent studies indicate that the use of oral doses of folic acid improves the treatment of antidepressants and anxiolytics [275]. The dose of folic acid should be 800 μg per day for at least two months [224].

In addition, these nutrients are also valuable for facilitating tissue repair and neuronal plasticity [276,277], as it is known that the nutrients, in combined action or individually, are essential for the improvement of intervention in mental mood disorders [31]. This is because antioxidant vitamins or vitamins D and B and other nutrients such as fatty acids of the N-3 and N-6 families function well as neurochemical modulators [278,279]. The synaptic connections and the functioning of neurotransmitters are essential for a well-balanced brain [280]. Adrenergic neurotransmitters such as dopamine, serotonin and noradrenaline are involved in mood disorders, and people with unbalanced levels of these neurotransmitters will favor the appearance of symptoms of depression and anxiety [281]. In fact, the pharmacology used in these patients works directly on increasing the transmission of some of these, depending on the type of medication chosen [4]. Therefore, when vitamin D deficiency is detected (values between 20–29 ng/mL), the antidepressive treatment can be supplemented with doses of 4000 international units or more, which enhances the effect of the pharmacology and reduces the risk of chronic depression [282,283]. A recent study reviewing the efficacy of treatment along this line verifies the benefits of using this supplement (specifically, 50,000 IU of vitamin D3 per week by mouth for 52 weeks). In these cases, depressive symptomatology values had decreased compared with the control group, which received a placebo [284].

Analyzing other interventions, we can find studies that have shown that the intake of more than 1.5 mg/day of omega-3 acids as a supplement to the normal diet significantly reduces the symptoms of depression [285]. Omega-3 is a lipid, specifically a fatty acid, that is present in our brain and contributes to the balance of the central nervous system. In this sense, a deficiency in levels of this lipid may increase the risk of suffering from various mental disorders, including mood disorders [286].

All this information is invaluable in understanding the mental disorder and how the drugs work on the neurotransmitters to be addressed by the medication [287]. Nutritional ergogenic therapy can make the difference between successful treatment and treatment failure [288]. In this regard, using polyunsaturated fatty acids and minerals will be essential for improving cognitive function. Similarly, the intake of antioxidants, vitamin C and vitamins of the B-family will improve the functioning of neurotransmitters and methylation reactions, improving memory and cognitive performance [4]. Vitamin D supplementation helps to improve depressive symptoms and reduce emotional lability, favoring behavioral stabilization in mood-related illnesses [289]. Recent studies show that applying a dose of 15 micrograms per day for at least 3 months, together with the prescribed antidepressant, significantly reduces scores on the Beck Depression Inventory applied in patients separated into an experimental group, receiving this daily dose of vitamin D, and a control group, not receiving this dose [290,291].

Folate is equally important for the central nervous system, and nutritional support of this supplement will help modulate mood by maintaining adequate levels of serotonin and catecholamine synthesis [292]. Previous studies indicate that low levels of folic acid are a risk factor for depression and other mood disorders [293,294]. Another important nutrient is selenium, which, among other functions, is an antioxidant that works in conjunction with other enzymes to ensure the proper functioning of the immune system and thyroid gland. In patients diagnosed with depression, those who are fed high doses of selenium report better recovery than those who do not have this nutrient [295,296]

The same applies to iron and zinc, as iron deficiency negatively affects the metabolic function of neurotransmitters and alters myelination and thyroid hormone assimilation. As for zinc, deficiencies in this nutrient cause immunosuppression, which often leads to symptoms of depression [297]. Therefore, a reduction in these nutrients will favor the development of behavioral abnormalities such as anxiety and depression [298]. Other studies have shown that low serum or red blood cell folate levels in red blood cells are associated with an increased risk of depression [299,300], as deficiencies in the B group of vitamins hinder cellular processes involved in the elimination of homocysteine, an amino acid directly related to irritable mood 303. Studies indicate that a dose of between 1000 and 3000 mg daily of vitamins B3 and B6, which promote serotonin production, reduces symptoms of depression and anxiety after 8 weeks [301,302]. Some studies have even reported doses of more than 10,000 mg daily in major depression [303,304].

The studies that are currently presenting significant results regarding the importance of using ergogenic aids to alleviate the effects of mood disorders give a new vision of how to approach these illnesses in a multidisciplinary way and reinforce the good results that can be obtained with pharmacology and psychotherapy alone [36,305]. In this line, and especially after the COVID-19 pandemic, showing the increase in stress factors related to the subsequent mental impact, the use of antioxidant and anti-inflammatory ergogenic aids would be important to plan a non-pharmacological intervention [15,306,307].

## 9. Multidisciplinary Interventions in Anxiety and Depression

The WHO reports that anxiety disorders are the most common mental disorders worldwide, with specific phobia, major depressive disorder, and social phobia being the most common anxiety disorders [172]—these are also considered causes of disability worldwide. It affects approximately 120 million people worldwide, and it is estimated that around 20% of non-diagnosed people will develop a depressive condition at least once in their lives, while 1 in 13 will suffer from anxiety. And yet, 75% of those people with mental disorders will remain untreated in developing countries, with nearly 1 million annual suicides [173].

The physiological view of depression is based on the deregulation of the concentration of neurotransmitters, and the usual treatment has consisted of psychological therapy and medication. Also, pharmacological therapies have increased significantly in the last two years due to the COVID-19 pandemic [174], although their benefits have been limited in the adult and geriatric population, even when compared with placebo [175]. Furthermore, psychological interventions such as cognitive behavioral therapy, one of the most used techniques in this type of pathology or interpersonal psychotherapy, have demonstrated effectiveness in improving depressive symptoms, with similar results to medication [176]. Along with these non-pharmacological therapies, physical exercise is added due to its excellent role in regulating dopaminergic discharge, autonomic regulation, and modulation in the production of serotonin, dopamine, glutamate, and gamma-aminobutyric acid, among others [17].

It is known that prolonged low physical activity has detrimental effects on health. Regular physical exercise helps to maintain physical and mental well-being, which facilitates an improvement in people’s quality of life. Different authors have proposed methodologies and associations between physical activity and improvement in mental health symptoms among various populations [177]. Benefits related to changes in mood and creativity may appear even after a single session [178]. The authors found that participation in school sports was a statistically significant predictor for milder depressive symptoms and better self-perceived mental health [179]. Physically active adolescents had higher self-esteem and life satisfaction, and a high level of physical activity was associated with a reduced likelihood of psychological distress among high school seniors [180]. In resume, the authors suggest that acute exercise responds better to acute than chronic anxiety. Yet, excessive physical activity may lead to overtraining and generate psychological distress and disorders. Controlled methodologies and studies are needed to clarify the mental health benefits of physical exercise, making classifications by age groups, which we will address below.

Various studies have investigated the beneficial effects of physical activity on the mental health of children and adolescents. Jewett et al. found that participation in school sports was a statistically significant predictor for milder depressive symptoms and better self-perceived mental health [181]. Doré et al. found in a 5-year follow-up study that a physical activity profile of ≥2 years was associated with better mental health compared to 0 years of participation [182]. Guddal et al., in a population-based follow-up study of Norwegian schoolchildren, found that physically active adolescents and team sports participants had higher self-esteem and life satisfaction, and a high level of physical activity was associated with a reduced likelihood of psychological distress among high school seniors [183]. Snedden et al. compared the role of physical activity on health-related quality of life among high school student-athletes and first-semester college students, finding that higher levels of sport and physical activity were associated with more positive mental health in these populations [184]. In a study that examined the relationship between motor competence, health-related physical fitness, and mental health outcomes in adolescents who played volleyball, soccer, and ultimate frisbee, an interaction of motor competence and health-related physical fitness was found with the mental health of adolescents [185].

Clemente studied the changes in depression, anxiety, happiness, sleep, motivation, and autonomic modulation before and after six multidisciplinary sessions of cognitive behavioral therapy, combined with aerobic physical activity and nutritional intervention, in a subject with a mixed anxiety and depression disorder. They found a reduction in the values of depression according to the HAM-D up to the classification of no depression, a reduction in the anxiety trait, and an increase in the subjective perceptions of sleep and happiness after the six intervention sessions [40].

In a study that evaluated the inverse relationship between inactivity and mental health in adults aged 20 to 88 years who completed a maximal treadmill fitness test and self-report measures of usual physical activity, depressive symptoms (Depression Scale of the Center of Epidemiological Studies; CES-D) and emotional well-being (General Well-Being Schedule; GWB), a significant inverse relationship was found between the maximum CR physical condition and the CES-D score (*p* < 0.0001), and a significant positive relationship between CR fitness and GWB score (*p* < 0.0001) [186]. Pozuelo Carrascosa et al. analyzed the relationship between resilience, cardiorespiratory fitness, and quality of life-related to mental health; the results showed that the values of quality of life related to mental health were significantly higher in students who had a good cardiorespiratory physical condition and a high level of resilience [187]. In a community study of men living in poor conditions in a Greek refugee camp, the frequency of participation in an 8-week exercise and sports training program appeared to have the potential to have a positive impact on the health of refugees, although the results were not conclusive [188]. In a clinical trial among 170 people with dementia living in nursing homes with an average age of 86 years, undergoing intensive strengthening and balance exercises in small groups twice a week for 12 weeks was significantly associated with improved strength after the intervention (*p* < 0.05) and apathy was reduced in patients with dementia [189]. Stella et al., in a controlled clinical trial, analyzed the effects of motor intervention on the neuropsychiatric symptoms of Alzheimer’s disease, finding that patients who received the intervention (aerobic exercises and functional balance exercises) presented a significant reduction in neuropsychiatric conditions compared to controls. Likewise, the burden and stress of the caregivers responsible for the patients who participated in the intervention decreased significantly compared to the controls [190]. Landi et al., in a study that addressed the impact of a moderate-intensity exercise program in frail, elderly, and demented patients living in nursing homes, a statistically significant reduction in behavioral problems (for example, wandering, physical abuse and speech, and sleep disorders) and a significant reduction in the use of antipsychotic and hypnotic medications [191]. Neville et al., in a study of elderly nursing home residents in Queensland, Australia, suggested that a dementia-specific aquatic exercise intervention reduces symptoms in people with dementia and improves psychological well-being in people with moderate to serious dementia [192]. Sampaio et al. and Christofoletti et al. found similar results in populations of institutionalized older adults diagnosed with dementia [193,194].

Physical activity is a protective factor against depression and anxiety in the general population and in young people at high risk of depression [40]. There is some evidence to support that high-, medium-, and even low-impact exercise interventions [for example, yoga and light stretching, moderately intense walking, cycling, or team sports] may be beneficial in improving symptomatology, cognition, and both positive and negative quality of life in schizophrenia [195]. Studies suggest that exercise may also be useful in reducing the side effects of antipsychotic medications, such as weight gain, which could improve adherence to pharmacological therapy for psychosis; however, more research is needed [196,197].

## 10. Practical Applications and Clinical Guidelines

This narrative review consolidates the current state of research on the relationship between nutrition, mental health, and related factors, particularly focusing on anxiety and depression. Based on this comprehensive examination of the scientific literature, several practical applications and clinical guidelines can be derived to help healthcare practitioners—including clinicians, psychologists, and psychiatrists—integrate nutritional considerations into their practice when addressing mental health disorders.

### 10.1. Holistic Approach to Mental Health Management

It is essential to approach the treatment of mental health disorders such as depression and anxiety from a multifactorial perspective. These disorders are not only influenced by genetics, socioeconomic status, and psychosocial factors but also by nutritional status, inflammation, gut microbiota, and dietary patterns. Given their complexity, a holistic management plan that includes both conventional therapies and nutritional interventions should be adopted.

### 10.2. Key Recommendations for Clinical Practice


Incorporating Nutritional Assessments in Mental Health Care: Routine assessments of patients’ dietary habits should be incorporated into clinical settings, especially for those diagnosed with anxiety and depressive disorders. Given that poor diet quality is a significant risk factor for mental health issues, identifying nutritional deficiencies or harmful eating patterns is crucial for crafting effective interventions.Dietary Interventions as Preventative and Therapeutic Measures: Specific dietary patterns, such as the Mediterranean diet, have been linked to reduced risks of depression and anxiety. Clinicians should advocate for diets rich in fruits, vegetables, whole grains, lean proteins, nuts, and essential fatty acids, particularly omega-3s, which are known to play a role in modulating neurotransmitter function. Additionally, diets low in saturated fats, refined sugars, and processed foods can help reduce inflammation and improve mental health outcomes.Addressing the Gut–Brain Axis: The gut microbiome is emerging as a central player in mental health, influencing inflammation, neurotransmitter production, and brain function. In clinical practice, supporting gut health through the promotion of prebiotic- and probiotic-rich diets can have positive effects on mental health outcomes. This may include foods such as yogurt, fermented vegetables, and fiber-rich fruits and vegetables.Supplementation and Ergogenic Aids: For patients with diagnosed deficiencies or specific risk factors, supplementation with key nutrients, such as vitamins B6, B12, D, folate, magnesium, zinc, and omega-3 fatty acids, should be considered. These nutrients support neurotransmitter synthesis and help regulate mood, inflammation, and oxidative stress. Clinicians should monitor supplementation carefully, ensuring appropriate dosages based on individual needs.Integration of Physical Activity and Nutritional Counseling: As studies show that physical activity positively influences mental health by regulating neurotransmitter levels and reducing anxiety, clinicians should recommend moderate-intensity exercise combined with proper nutrition. Programs that integrate physical activity, nutritional counseling, and cognitive-behavioral therapy (CBT) can lead to significant improvements in anxiety and depression.


### 10.3. Psychobiotics and Emerging Therapies

Recent research highlights the potential of psychobiotics—probiotics that influence mental health through the gut–brain axis—as a novel therapeutic option. While still in its early stages, psychobiotic therapy, combined with dietary modifications, offers promising avenues for treating mental health disorders. Clinicians should stay updated on developments in this area to consider such interventions when scientifically validated.

### 10.4. Addressing Lifestyle Factors and Allostatic Load

Given that modern Western lifestyles, characterized by high caloric intake and chronic stress, contribute to increased allostatic load (the physiological wear and tear due to stress), healthcare providers must encourage lifestyle modifications. These include stress management techniques, regular physical activity, and dietary improvements to reduce inflammation and support mental well-being.

### 10.5. Personalized Nutrition in Mental Health

Personalized approaches to nutrition are crucial, given that individual differences in genetics, metabolism, and environmental factors can significantly influence how dietary interventions impact mental health. Clinicians should work closely with nutritionists and dietitians to develop individualized nutrition plans that address the specific needs of patients.

## 11. Conclusions

The results show how there is a direct relationship between what we eat and the state of our nervous system. The gut–brain axis is a complex system in which the intestinal microbiota communicate directly with our nervous system and provide neurotransmitters for proper functioning. An imbalance in our microbiota due to poor nutrition will cause an inflammatory response that, if sustained over time and together with other factors, can lead to disorders such as anxiety and depression. Modifications in nutritional behaviors and the use of ergonutricional components are presented as important non-pharmacological interventions in anxiety and depression prevention and treatment.

## Figures and Tables

**Figure 1 metabolites-14-00549-f001:**
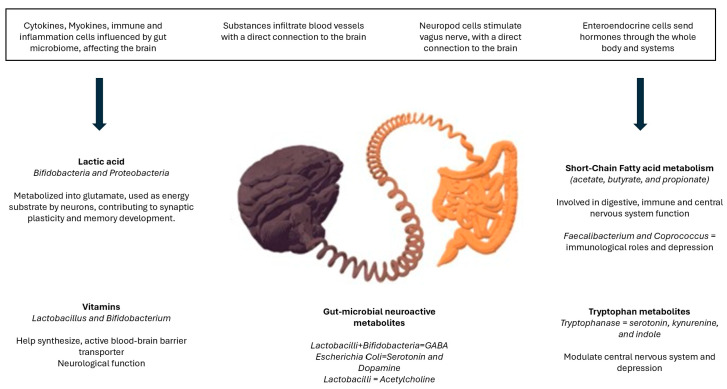
Gut–brain axis behavior and influence on mental health.

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
