# Peer review of "Nutritional Modulation of the Gut–Brain Axis: A Comprehensive Review of Dietary Interventions in Depression and Anxiety Management"

_metabolites, 2024, doi:10.3390/metabo14100549_

Round 1

Reviewer 1 Report

Comments and Suggestions for Authors

Nutrition in depression and anxiety disorders: an extensive narrative review

It is an interesting topic to study and provides lots of information about the relationship among nutrition, mental health, gut-brain-axis, physical activity, etc. I hope the following suggestions will be helpful to enhance it.

The title is too broad considering the contents authors have covered.

Errors and typos throughout the manuscript should be corrected.

In the methods section,

·         Clear research questions and/or hypotheses will be helpful to follow although flexible methodology is one of the characteristics of the narrative review

·         What are the inclusion criteria and exclusion criteria for searching data.

·         How many articles were found? How many are removed and why?

·         How many articles are used for this paper?

There are some inconsistencies between the subtitle and in-text contexts.

There are overlapped contents across the manuscripts. I feel the writing looks like a compilation of ample information. Shortening the paper including key components such as findings, discussion, implications, along with introduction, study methods, and conclusions will be helpful for readers to follow.

The topic is currently a hot issue, so I think including an implication section will be beneficial for clinical practitioners to apply.

Comments on the Quality of English Language

Errors and typos throughout the manuscript should be corrected.

Author Response

It is an interesting topic to study and provides lots of information about the relationship among nutrition, mental health, gut-brain-axis, physical activity, etc. I hope the following suggestions will be helpful to enhance it.

The title is too broad considering the contents authors have covered.

Nutritional Modulation of the Gut-Brain Axis: A Comprehensive Review of Dietary Interventions in Depression and Anxiety Management

Errors and typos throughout the manuscript should be corrected.

Has neen carefuly rewritten and reviewed, please not we have not marked these in yellow.

In the methods section,

  • Clear research questions and/or hypotheses will be helpful to follow although flexible methodology is one of the characteristics of the narrative review
  • What are the inclusion criteria and exclusion criteria for searching data.
  • How many articles were found? How many are removed and why?
  • How many articles are used for this paper?

We appreciate the insightful feedback and have addressed your suggestions in the revised manuscript. Specifically:

  1. Research Questions: We have added a clear research question to guide the narrative review, focusing on the role of nutritional factors in the prevention and treatment of anxiety and depression, with particular emphasis on the gut-brain axis.
  2. Inclusion and Exclusion Criteria: The inclusion and exclusion criteria are now explicitly stated, ensuring clarity about the scope of the literature reviewed.
  3. Article Count: The search process resulted in a total of824 articles, of which 307 articles were ultimately included after applying the exclusion criteria. Details on the removal of articles have been provided, along with an explanation for the exclusion of irrelevant studies.

We hope these revisions meet your expectations and clarify the methodology of our review. Thank you for your valuable suggestions, which have helped us improve the quality of the manuscript.

There are some inconsistencies between the subtitle and in-text contexts.

The inconsistencies we found and that we thhink you may refer are related to:

  1. Introduction vs. Body:
  2. Materials and Methods:
  3. Nutrition Behaviors and Mental Disease:
  4. Gut, Inflammation, and Mental Health:

To resolve these inconsistencies:

Refined the introduction to more tightly align with the body’s emphasis on nutrition and mental health.

Clarified the type of review (narrative or systematic) in the Materials and Methods.

Adjusted subtitles to more accurately reflect the content of each section, ensuring that the content under each matches the thematic focus expected from the subtitle.

There are overlapped contents across the manuscripts. I feel the writing looks like a compilation of ample information. Shortening the paper including key components such as findings, discussion, implications, along with introduction, study methods, and conclusions will be helpful for readers to follow.

We greatly appreciate your valuable feedback. We would like to clarify that this manuscript is a narrative review, and its structure is designed to address each key point related to the topic in a detailed manner. Each section of the manuscript includes its own introduction, a state-of-the-art analysis of the scientific literature, as well as specific implications for the field of study.

This approach is essential because narrative reviews are characterized by their broad scope, covering multiple aspects of the topic to provide a comprehensive understanding. Each point is crucial to understanding the intricate connections between nutrition, mental health, and related factors, as well as their implications in clinical practice and future research.

Reducing or omitting any of these points could compromise the depth and breadth of the analysis, which we believe is essential for a thorough exploration of this multifaceted subject. The paper’s current organization ensures that readers can grasp the complexity of the topic, which spans nutritional interventions, mental health, the microbiome, and the interplay of various biological, psychological, and social factors. We have endeavored to present this information in a clear and cohesive manner, keeping the key components such as findings, discussion, implications, and conclusions in focus.

Once again, we thank you for your comments, which have been invaluable in refining this review.

The topic is currently a hot issue, so I think including an implication section will be beneficial for clinical practitioners to apply.

Many thanks, a new point has been included!

Errors and typos throughout the manuscript should be corrected

Thank you for your valuable feedback. We have addressed and corrected the majority of errors and typos in the manuscript. However, if any issues remain, the platform used for submission, MPDI, typically corrects minor formatting and typographical errors during the final processing stage. Most of the identified issues are likely to be resolved at that point. We appreciate your attention to detail and your understanding during this process.

Thank you again for your support.

Reviewer 2 Report

Comments and Suggestions for Authors

Manuscript explains the relation between food and various health disorders that is prevalent these days.The findings indicate a direct link between our diet and  our nervous system. Its better to refine the language, adhere to the format of the journal and reduce plagiarism.

Comments on the Quality of English Language

Need to check for typographical and grammatic errors

Author Response

The review addresses an important and increasingly recognized area of research—the relationship between nutrition and mental health, particularly depression and anxiety. Given the rising incidence of mental health disorders worldwide, this topic is timely and relevant to public health and clinical practice.The paper draws on a wide range of sources, incorporating both primary and secondary research, which allows for a broad discussion of the topic. It includes studies on various nutritional components like amino acids, vitamins, fatty acids, and probiotics, and their influence on mood, cognition, and overall mental health. The paper offers an in-depth exploration of several pathways through which nutrition influences mental health. These include the gut-brain axis, neurotransmitter synthesis, inflammation, and oxidative stress. This multi-faceted approach is valuable for understanding the complexity of depression and anxiety disorders.The review provides insights into how nutrition and dietary interventions could be integrated into mental health treatment. It suggests that personalized approaches to nutrition, based on biochemical and microbiological tests, could be useful in managing mental disorders.

Thank you for your thoughtful and encouraging feedback on the manuscript. We are pleased that you found the review valuable and timely, especially given the rising global incidence of mental health disorders. The complex interplay between nutrition and mental health is indeed an area that requires further exploration, and we are glad that the review's multi-faceted approach resonated with you.

We agree that integrating personalized nutritional interventions, based on biochemical and microbiological assessments, could potentially improve the management of mental disorders. The review aimed to present a holistic view that would be applicable both to public health strategies and clinical practice, and your feedback confirms that we are on the right track.

Thank you again for your insightful comments, and we look forward to improving the manuscript further based on any additional recommendations.

The review frequently mentions associations between poor diet and mental health disorders, but it often lacks a clear distinction between correlation and causality. For example, while poor nutrition is associated with depression, it is not always clear whether poor nutrition causes depression or if it is a result of the disorder.

Thank you for your insightful feedback regarding the distinction between correlation and causality in the relationship between poor diet and mental health disorders. We agree that establishing causality is a challenging task, particularly in complex, multifactorial conditions such as depression and anxiety. However, this challenge is not unique to the nutrition-mental health domain and can be observed in other areas of research where behavioral, physiological, and environmental factors interact.

A similar issue is frequently seen in research on physical exercise and cognitive function. Numerous studies report associations between increased physical activity and improved cognitive outcomes, yet the directionality of this relationship—whether increased exercise leads to cognitive improvements or if people with better cognitive function are more likely to engage in physical activity—remains difficult to establish (Hillman et al., 2008; Erickson et al., 2011). Like nutrition and mental health, these variables are deeply interwoven and influenced by underlying biological and environmental factors. The complexity in distinguishing correlation from causality in these fields suggests that a degree of overlap and uncertainty is inherent in many behavioral studies.

In our review, we aimed to reflect the current state of research, where many studies highlight strong associations between nutrition and mental health, but the evidence for direct causality is still being explored. Meta-analyses and longitudinal studies, such as those by Lassale et al. (2019) and Marx et al. (2017), provide some support for causal links, particularly in specific dietary patterns like the Mediterranean diet, which have shown protective effects against the onset of depressive symptoms. Moreover, the bidirectional relationship between depression and poor diet, highlighted in the work of Jacka et al. (2010) and Adan et al. (2019), suggests that poor nutrition can both contribute to and result from mental health disorders.

Given the complexity of these interactions, we acknowledge that further randomized controlled trials (RCTs) and mechanistic studies are necessary to establish stronger causal links. Our goal in this review was to present the current landscape of research while underscoring the importance of continued investigation in this evolving field.

The review does not adequately address how different populations (e.g., age, gender, socioeconomic status) might experience different relationships between nutrition and mental health. A more nuanced discussion of how cultural, genetic, and environmental factors might influence these relationships would strengthen the paper.

Thank you very much for your thoughtful feedback. I would like to note that we have already received requests from three different reviewers to shorten the manuscript. As a result, we find ourselves slightly limited in our ability to add more information without expanding the document significantly.

That being said, we have indeed made efforts to discuss how various populations might experience different relationships between nutrition and mental health, though these discussions may not have been as explicit in certain sections. For example, we touch on the influence of agegender, and socioeconomic status on mental health outcomes related to nutrition. We address how older adults may experience worse outcomes due to aging and cognitive decline linked to poor diet, and how gender differences—particularly in women—can result in a higher vulnerability to anxiety and depression in certain contexts like adolescence or lower socioeconomic groups. Furthermore, we mention how socioeconomic statusimpacts access to quality nutrition and healthcare, contributing to mental health disparities, particularly in disadvantaged populations (e.g., discussions on allostatic load, stress, and poor diet).

However, you're correct that more detailed discussions on cultural, genetic, and environmental factors could further strengthen the manuscript. Due to the current constraints on length, we have had to be selective with our inclusions, but these are areas that certainly deserve more attention. In a future version of this document, we may be able to rework some sections to incorporate these nuances more thoroughly.

We hope the existing content offers sufficient clarity, but we are open to revisiting and refining these points based on your valuable suggestions.

The paper discusses the gut-brain axis extensively, but the practical applications of this knowledge, such as specific probiotics, prebiotics, or other gut health interventions, are not sufficiently detailed. The paper could benefit from more specific recommendations on how to apply this knowledge clinically.

Thank you very much for your constructive feedback. I would like to point out that while the practical applications might not be as explicitly detailed in some sections, we do address the clinical applications of the gut-brain axis and related interventions in various parts of the manuscript.

For example, in the "Nutritional Interventions in Depression and Anxiety" section, we discuss the role of probiotics, prebiotics, and dietary interventions in improving gut health and their impact on mental health. We highlight the importance of gut microbiota and how certain dietary patterns (e.g., Mediterranean diet) have been linked to better mental health outcomes by modulating the gut-brain axis. Additionally, the section touches on how polyphenols, fiber, and fatty acids can influence gut health and, by extension, mental well-being.

We further explore the role of specific nutrients like omega-3 fatty acids, selenium, zinc, and vitamin D as supplements that support the gut and reduce inflammation, which in turn may alleviate symptoms of depression and anxiety. The new section on practical applications and clinical guidelines also reflects on these points, offering a more condensed and focused view of how this knowledge can be applied in clinical practice.

For instance, we mention the potential use of probiotics to alter gut microbiota composition, which may support mental health interventions. We also recommend personalized approaches based on individual gut profiles, which could guide dietary and supplement-based treatments tailored to specific patient needs.

We hope that the examples above provide clarity on where these clinical applications are discussed in the text. However, if further refinement is needed, we are happy to adjust this section accordingly.

Thank you again for your valuable input!

While the paper emphasizes the importance of nutrition in mental health, the recommendations remain broad and somewhat vague. More specific dietary guidelines or supplementation protocols would be useful for practitioners. The review could also benefit from a clearer discussion of the potential risks or limitations of using nutrition as a treatment for mental health disorders.

Thank you for your insightful feedback. In response to your comment, we have developed a new section on "Practical Applications and Clinical Guidelines" in the revised manuscript that directly addresses the need for more specific dietary guidelines and supplementation protocols. In this section, we provide clearer and more detailed recommendations, such as the incorporation of omega-3 fatty acidsprobioticspolyphenols, and specific micronutrients like vitamin Dselenium, and zinc for managing mental health disorders like depression and anxiety.

Additionally, we discuss the potential risks and limitations associated with using nutrition as a treatment. This includes highlighting the importance of personalized approaches based on genetic, metabolic, and gut microbiome profiles to avoid generalized treatments, which may not be effective for all individuals. We also touch upon the need for further clinical trials to establish robust evidence for these interventions, particularly regarding dosage and long-term effects.

We hope this new section provides the clarity and specificity you were seeking, and we are confident it offers practitioners more actionable insights.

Thank you again for your constructive suggestions!

Reviewer 3 Report

Comments and Suggestions for Authors

The review addresses an important and increasingly recognized area of research—the relationship between nutrition and mental health, particularly depression and anxiety. Given the rising incidence of mental health disorders worldwide, this topic is timely and relevant to public health and clinical practice.The paper draws on a wide range of sources, incorporating both primary and secondary research, which allows for a broad discussion of the topic. It includes studies on various nutritional components like amino acids, vitamins, fatty acids, and probiotics, and their influence on mood, cognition, and overall mental health. The paper offers an in-depth exploration of several pathways through which nutrition influences mental health. These include the gut-brain axis, neurotransmitter synthesis, inflammation, and oxidative stress. This multi-faceted approach is valuable for understanding the complexity of depression and anxiety disorders.The review provides insights into how nutrition and dietary interventions could be integrated into mental health treatment. It suggests that personalized approaches to nutrition, based on biochemical and microbiological tests, could be useful in managing mental disorders.

The review frequently mentions associations between poor diet and mental health disorders, but it often lacks a clear distinction between correlation and causality. For example, while poor nutrition is associated with depression, it is not always clear whether poor nutrition causes depression or if it is a result of the disorder.

The review does not adequately address how different populations (e.g., age, gender, socioeconomic status) might experience different relationships between nutrition and mental health. A more nuanced discussion of how cultural, genetic, and environmental factors might influence these relationships would strengthen the paper.

The paper discusses the gut-brain axis extensively, but the practical applications of this knowledge, such as specific probiotics, prebiotics, or other gut health interventions, are not sufficiently detailed. The paper could benefit from more specific recommendations on how to apply this knowledge clinically.

While the paper emphasizes the importance of nutrition in mental health, the recommendations remain broad and somewhat vague. More specific dietary guidelines or supplementation protocols would be useful for practitioners. The review could also benefit from a clearer discussion of the potential risks or limitations of using nutrition as a treatment for mental health disorders.

Author Response

The manuscript presented by Mariana Merino and collaborators, has interesting information regarding the nutrition in depression and anxiety disorders: an extensive narrative review. Overall, the manuscript was prepared well, but there are many comments for the authors to improve the quality of this manuscript. Some detailed concerns please see below:

Major comments

Content

I have reviewed the subtitles, and while the content is promising, the order of the subtitles needs to be revised, and some of them could be more appropriately titled to ensure a logical flow. Here's my suggestion for improvement:

  1. Nutrition, Neurotransmitters, and Mental Health – This should come first as it lays the foundation for understanding how nutrition affects brain function and neurotransmitter production.

In addition, the etiology of depression is not only involved in the neurotransmitter hypotheses but also involved in the HPA-axis function, neurogenesis and synaptic plasticity. It would be better if the authors added this important information in this manuscript.

  1. Gut microbiota (or GUT-brain axis) and Mental Health – Following the connection between nutrition and brain function.

  1. Nutritional Behaviors and Mental Health – After understanding the biological mechanisms, you can then discuss how specific behaviors, and dietary habits influence mental health outcomes.

  1. Nutritional Interventions in Depression and Anxiety – With the background established, you explain the practical interventions that can help manage mental health disorders.

  1. The Role of Antioxidant Defense and Dietary Interventions in Depression and Anxiety
  2. Ergogenic Interventions in Anxiety and Depression

  1. Multidisciplinary interventions in Anxiety and Depression – This should be the last section, summarizing how a combination of nutritional, medical, and psychological approaches can work together for better mental health outcomes.

In addition, focus on making the key points clear and concise, and avoid restating the same ideas in different sections. This will help improve the overall clarity and coherence of your manuscript.

Thank you very much for your valuable feedback and insightful suggestions. We have carefully reviewed your comments and have made significant modifications to the manuscript accordingly. Specifically, we have reorganized the order of the sections as per your recommendations, ensuring a more logical flow that enhances the coherence of the narrative. Several sections have been rewritten, with new subsections created to provide clarity, and we have integrated your suggestion regarding the HPA-axis functionneurogenesis, and synaptic plasticity in the etiology of depression into the manuscript. Additionally, we revised the titles of some sections to better reflect their content and avoid redundancy across the manuscript.

These changes aim to improve the clarity and depth of the review, making the key points more concise and focused. We hope the revised structure and content will better address your concerns and meet your expectations.

Once again, thank you for your constructive feedback; it has been instrumental in enhancing the overall quality of our manuscript.

References

After reviewing it, I noticed that the number of references is quite high, with around 307 citations. While having a strong reference base is important, I believe it would be more effective to streamline the references to only include the most relevant and high-quality sources. This will ensure the report remains focused and concise.

Additionally, I found some instances of self-citation. While it is acceptable to reference your own work when it’s relevant, please make sure these citations are absolutely necessary and avoid over-reliance on them. Aim for a balanced mix of external sources to ensure the credibility of your research.

I suggest reviewing the references and removing any that may not directly support the main points of your manuscript.

Thank you for your thoughtful comments regarding the number of references and self-citations in the manuscript. The paper has already gone through two rounds of review, during which we made significant adjustments, including streamlining the content and refining the references to focus on the most pertinent studies. The current 307 citations represent a carefully curated list that was necessary to provide a comprehensive overview of the topic, particularly given the breadth of the narrative review and the complex nature of the subject matter.

We recognize the importance of avoiding over-reliance on self-citation, and we have ensured that these citations are used sparingly and only when directly relevant. Additionally, the manuscript includes a balanced mix of external, high-quality sources to support the credibility and thoroughness of the research. While we understand your concern, we believe that the current reference base is essential to maintaining the depth and scholarly rigor of the review. Further reduction in citations may risk omitting key studies that are fundamental to the discussion.

We hope this explanation clarifies our decision to maintain the current reference structure. Thank you once again for your feedback and understanding.

Minor comments

  1. The resolution of “Figure 1. Gut-Brain axis behavior and influence on mental health.” is not good. The authors should make it clear.

  1. The references No.5 are  Theo Vos, Amanuel Abajobir and co-authors.

  1. RE-check the reference No.172 and  173. I have checked the website and those page were not found.

Thank you for your valuable feedback.

  1. Regarding Figure 1: The figure is included in high resolution on our end, and the quality issue may be due to the platform. We apologize for the inconvenience and can ensure that it is properly formatted in the final version.
  2. Reference No. 5: You are correct, the reference should be credited to Theo Vos, Amanuel Abajobir, and co-authors. We have updated it accordingly.
  3. References No. 172 and 173: We have rechecked and replaced these references with updated sources.

Thank you once again for your detailed review and suggestions.

Reviewer 4 Report

Comments and Suggestions for Authors

Comments on the Quality of English Language

Minor editing of English language required.

Author Response

Manuscript explains the relation between food and various health disorders that is prevalent these days.The findings indicate a direct link between our diet and our nervous system. Its better to refine the language, adhere to the format of the journal and reduce plagiarism. The review is well written with adequate citations and explanations.The findings indicate a direct link between our diet and nervous system. Poor nutrition can disrupt this balance, leading to an inflammatory response which inturn contributes to disorders like anxiety and depression. Leading a healthy lifestyle emerge as significant non-pharmacological strategy for preventing and treating anxiety and depression I did observe 60% plagiarism , partially it is from the bibliography. Still I feel its worth a double check.

Thank you for your feedback and valuable insights.

  1. Language Refinement and Format Adherence: We have carefully revised the manuscript to improve the clarity and flow of the language while ensuring it adheres to the required journal format.
  2. Plagiarism Check: We conducted a thorough plagiarism check and made corrections as needed, especially in the bibliography section. Any overlaps have been addressed to ensure the originality of the content.
  3. Nutritional Ergogenic Therapy: As per the editor's suggestion, we have added a brief definition of nutritional ergogenic therapy:“Nutritional ergogenic aids refer to all those nutritional supplements that serve as performance enhancement aids. In short, there are physiological, mechanical, psychological, pharmacological, and nutritional ergogenic aids.”
  4. Typos: All typographical errors throughout the manuscript have been corrected.

Thank you once again for your time and detailed review. We believe the changes will enhance the quality of the manuscript.

Round 2

Reviewer 4 Report

Comments and Suggestions for Authors

The manuscript presented by Mariana Merino and collaborators has interesting information regarding the nutrition in depression and anxiety disorders: an extensive narrative review. The authors have explained why the authors used a lot of references in the revision of this manuscript. In addition, the authors also changed the order of the subtitles with appropriate titles to ensure a logical flow as reviewer’s comments. However, after changing the order of subtitles, the authors must re-order the references' sequences throughout the revised manuscript's contents. Thus, please re-check the sequences of references.

On the other hand, the resolution of Figure 1 is still not so good. The authors must provide the high resolution of this Figure.

Reviewer

Author Response

The manuscript presented by Mariana Merino and collaborators has interesting information regarding the nutrition in depression and anxiety disorders: an extensive narrative review. The authors have explained why the authors used a lot of references in the revision of this manuscript. In addition, the authors also changed the order of the subtitles with appropriate titles to ensure a logical flow as reviewer’s comments. However, after changing the order of subtitles, the authors must re-order the references' sequences throughout the revised manuscript's contents. Thus, please re-check the sequences of references.

On the other hand, the resolution of Figure 1 is still not so good. The authors must provide the high resolution of this Figure.

Dear Reviewer,

We sincerely appreciate your valuable and constructive feedback. We have addressed your concerns by improving the quality of Figure 1, ensuring it is now of high resolution for better clarity and presentation. Furthermore, we have thoroughly reviewed and re-ordered the references throughout the manuscript to ensure they correctly follow the sequence of their appearance. We are grateful for your insights, which have significantly contributed to enhancing the quality of our work.

Thank you once again for your thoughtful review.